# SEMANTIC HASHING WITH LOCALITY SENSITIVE EMBEDDINGS

## ABSTRACT

Semantic hashing methods have been explored for learning transformations into binary vector spaces. These learned binary representations may then be used in hashing based retrieval methods, typically by retrieving all neighboring elements in the Hamming ball with radius 1 or 2. Prior studies focus on tasks with a few dozen to a few hundred semantic categories at most, and it is not currently well known how these methods scale to domains with richer semantic structure. In this study, we focus on learning embeddings for the use in exact hashing retrieval, where Approximate Nearest Neighbor search comprises of a simple table lookup. We propose similarity learning methods in which the optimized base similarity is the angular similarity (the probability of collision under SimHash.) We demonstrate the benefits of these embeddings on a variety of domains, including a coocurrence modelling task on a large scale text corpus; the rich structure of which cannot be handled by a few hundred semantic groups.

## 1 INTRODUCTION

One of most challenging aspects in many Information Retrieval (IR) systems is the discovery and identification of the nearest neighbors of a query element in an vector space. This is typically solved using Approximate Nearest Neighbors (ANN) methods as exact solutions typically do not scale well with the dimension of the vector space. ANN methods typically fall into one of three categories: space partitioning trees, such as the kd-tree (Bentley (1975); Friedman et al. (1977); Arya et al. (1998)), neighborhood graph search (Chen et al. (2018); Iwasaki & Miyazaki (2018)) or Locality Sensitive Hashing (LSH) methods (Charikar (2002); Gionis et al. (1999); Lv et al. (2007)).

Despite their theoretical, intuitive, and computational appeal, LSH methods are not as prevalent in modern IR systems as are space-partitioning trees or neighborhood graph methods (Bernhardsson (2013); Chen et al. (2018); Johnson et al. (2017); Iwasaki & Miyazaki (2018)). Empirical studies demonstrate that LSH techniques frequently do not attain the same level of quality as space-partitioning trees (Muja & Lowe (2009)).

Nonetheless, space-partitioning and neighborhood graph search methods are expensive, both in data structure construction and in query time, and remain a bottleneck in many modern IR pipelines. As many modern retrieval tasks revolve around solving ANN for vector representations learned from raw, structured data, one might attempt to learn representations which are more suited towards efficient retrieval. Metric learning methods (Xing et al. (2003); Weinberger et al. (2006); Chechik et al. (2010); Hoffer & Ailon (2015); Kulis et al. (2013)) have been proposed for learning linear and non-linear transformations of given representations for improved clustering and retrieval quality. A class of related methods, semantic hashing or hash learning methods (Salakhutdinov & Hinton (2009)), have also been explored for learning transformations into binary vector spaces. These learned binary representations may then be used in hashing based retrieval methods, typically by retrieving all neighboring elements in the Hamming ball with radius 1 or 2.

Exact hashing retrieval algorithms, that is, Hamming ball "search" with radius 0, have a particular computational appeal in that search data structures are not needed nor is enumeration of all codes within a Hamming ball. In addition, binary representations that are suitable for exact hashing retrieval can also be used to identify groups of related items that can be interpreted as clusters in the traditional sense. As the number of clusters discovered by the algorithm isn't explicitly controlled

(only bounded by $2^d$,) algorithms generating binary embeddings suitable for exact hashing retrieval can be viewed as nonparametric clustering methods.

To this end, we propose a method for learning continuous representations in which the optimized similarity is the *angular similarity*. The angular similarity corresponds to the collision probability of SimHash, a hyperplane based LSH function (Charikar (2002)). Angular distance gives a sharp topology on the embedding space which encourages similar objects have nearly identical embeddings suitable for exact hashing retrieval.

Related work on similarity learning, LSH, and hash learning can be found in Section 2. The proposed models are found in Section 3. The experimental results, and other technical details, can be found in Sections 4. Finally, we conclude in Section 5.

## 2    PRELIMINARIES

### 2.1    SIMILARITY MODELLING

Similarity learning methods are a class of techniques for learning a similarity function between objects. One successful approach for similarity learning are "twin network" or "two tower architecture" models, in which two neural network architectures are joined to produce a similarity prediction (Bromley et al. (1994); Chopra et al. (2005); Huang et al. (2013)). The weights of these networks may be shared or not, depending on whether the two input domains are equivalent or not.

Let $i \in \mathcal{U}$ and $j \in \mathcal{V}$ be the identities of two objects, where $\mathcal{U}$ and $\mathcal{V}$ are the two domains across which a similarity function is to be learned. Let $\phi_u(i)$ and $\phi_v(j)$ be the input representations for the objects (these functions $\phi$ may be identity functions if the input domains are discrete.) These representations are then transformed through parameterized vector-valued functions $f_u(\cdot|\theta_u)$ and $f_v(\cdot|\theta_v)$, whose output are typically the learned representations $u_i = f_u(\phi_u(i)|\theta_u)$ and $v_j = f_v(\phi_v(j)|\theta_v)$.

A loss is then defined using pairwise labels $y_{ij}$ and an interaction function $s(u_i, v_j)$ which denotes the similarity or relevancy of the pair. Taking $f_u$ to be a mapping for each index $i$ to an independent parameter vector $u_i$ (similarly for $f_v$ and $v_i$), and taking $s(u_i, v_j) = u_i^T v_j$ with an appropriate loss results in a variety of matrix factorization approaches (Koren et al. (2009); Lee & Seung (2001); Mnih & Salakhutdinov (2008); Blei et al. (2003); Rendle et al. (2012); Pennington et al. (2014)).

Taking $f_u$ to be a neural network mapping a context $\phi_u(i)$ to a representation $u_i$ allows for similarity models that readily make use of complex contextual information. Common choices for the similarity function include transformations of Euclidean distance (Chopra et al. (2005)), and cosine similarity: $s(u_i, v_j) = \frac{u_i^T v_j}{||u_i|| ||v_j||}$ (Huang et al. (2013)). In addition, the loss can be defined for pairs (Chopra et al. (2005)), triplets (one positive pair, one negative pair) (Rendle et al. (2012); Chechik et al. (2010)), or on larger sets (Huang et al. (2013)).

### 2.2    LOCALITY SENSITIVE HASHING AND ANGULAR SIMILARITY

A Locality Sensitive Hash (LSH) family $\mathcal{F}$ is a distribution of hashes $h$ on a collection of objects $Q$ such that for $q_i, q_j \in Q$, (Indyk & Motwani (1998); Gionis et al. (1999); Charikar (2002))

$$Pr[h(q_i) = h(q_j)] = s(q_i, q_j) \tag{1}$$

for some similarity function $s$ on the objects. SimHash (Charikar (2002)) is a LSH technique developed for document deduplication but may be used in other contexts. For a vector representations $q \in \mathbb{R}^d$, SimHash draws a random matrix $Z \in \mathbb{R}^{d \times M}$ with standard Normal entries. The hash $h(q_i) \in \{0, 1\}^M$ is then constructed as

$$h(q_i)_m = \mathbb{1}[q_i^T Z_{:m} > 0]. \tag{2}$$

Intuitively, SimHash draws random hyperplanes intersecting the origin to separate points. A useful property of this hash function, as stated in Charikar (2002), is that

$$\psi(q_i, q_j) := Pr[h(q_i)_m = h(q_j)_m] = 1 - \frac{1}{\pi} \cos^{-1} \left( \frac{q_i^T q_j}{||q_i|| ||q_j||} \right),$$

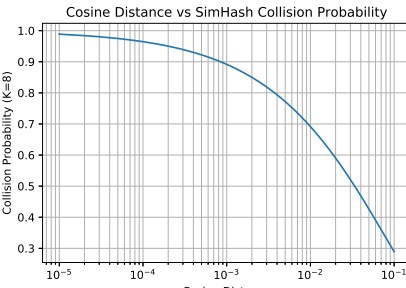 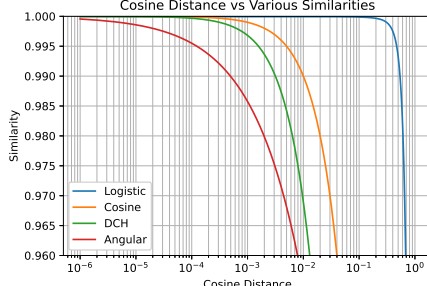

Figure 1: **(left)** Response of 8-bit SimHash collision probability vs. cosine distance. The plot indicates that vectors that are extremely close in cosine distance may not collide under SimHash. For example, a cosine distance of $0.001$ corresponds to a collision probability of only $0.9$. **(right)** Response of various similarities vs cosine distance. Angular distance (1-bit Simhash collision probability) induces the sharpest topology. The DCH distance uses $\gamma = 5$ and $d = 32$.

where the above probability is measured with respect to $Z$. $\psi(q_i, q_j)$, the collision probability for two vectors, is also known as the *angular similarity*, and $\xi = 1 - \psi$ is the *angular distance*, which is a proper metric (unlike the cosine distance $1 - \frac{q_i^T q_j}{||q_i||||q_j||}$). As the columns of $Z$ are independent, the collision probability for a $K$ bit hash is $\psi^K$.

## 2.3 LEARNING TO HASH

A related approach to similarity learning is hash learning methods, introduced in Salakhutdinov & Hinton (2009). These methods train binary embeddings directly and then use hash collisions or Hamming Ball search to retrieve approximate nearest neighbors. Binary representations lead to some technical challenges; Salakhutdinov & Hinton (2009) uses contrastive divergence for training, whereas Hubara et al. (2016) implement binary threshold activation functions with stochastic neurons.

Another approach (and the one followed in this work) is to avoid explicit binary representations in training and to introduce an quantization loss to penalize embeddings that are not close to binary, and to subsequently threshold these near-binary embeddings to binary ones. This type of quantization loss is distinct from those used in vector quantization methods (Ahalt et al. (1990); Kohonen (1990); Sato & Yamada (1996)) in which the data representations are fixed and the codes are learned; here the codes are fixed and the representations are learned.

The quantization loss introduced in Deep Hashing Networks (DHN) Zhu et al. (2016) is of the form

$$b(u_i|\theta) = \sum_d \log \cosh \left( |u_{id}| - 1 \right) \approx \left\| |u_i| - \mathbf{1} \right\|_1 . \tag{3}$$

Other quantization losses based on distances to binary codes have been used in Li et al. (2016); Liu et al. (2016). Cao et al. (2017) utilizes a quantization loss whose strength increases over time. Finally, Deep Cauchy Hashing (DCH) (Cao et al. (2018)) has shown improvements by utilizing a heavy-tailed similarity function with a similarly inspired quantization loss.

## 3 LOCALITY SENSITIVE EMBEDDINGS

Many similarity learning methods utilize dot products or cosine similarity to relate the embeddings of a pair to each other. For example GloVe (Pennington et al. (2014)) minimizes the weighted error between the dot product of the embeddings and a log-coocurrence matrix, and the DSSM model (Huang et al. (2013)) utilizes cosine similarity as the "crossing" layer between the two halves of a twin network. In general, embeddings trained in this way are not suitable for SimHash retrieval, as can be seen in Figure 1. If models are trained so as to minimize the error of a prediction made by cosine similarity, extremely low tolerances are required in order to achieve embeddings with

significant collision probability. Similar observations on the misspecification of cosine distance for Semantic Hashing were made in Cao et al. (2018). In this section, we define models in which collision probabilities of learned representations are directly optimized.

## 3.1 LOSS DEFINITION

In the following, we define a population loss through a data distribution $\mathcal{D}$ of relevant and irrelevant pairs. Each sample from $\mathcal{D}$ is a tuple $(y, i, j) \in \{0, 1\} \times \mathcal{U} \times \mathcal{V}$, where $\mathcal{U}$ and $\mathcal{V}$ are the sets across which a similarity is to be learned – for example, "users" and "items" in a recommender system. $y$ is the relevancy of the pair $(i, j)$.

The population losses we consider are expectations over $\mathcal{D}$ of a per-tuple loss $l$ with regularization terms $r$ per item:

$$L(\theta) = \mathop{\mathbb{E}}_{y,i,j \sim \mathcal{D}} l(y, i, j | \theta) + \lambda r(i | \theta) + \lambda r(j | \theta). \tag{4}$$

In practice, we minimize the empirical loss $\hat{L}$ constructed from a finite sample from $\mathcal{D}$, and we use $r(i|\theta) = b(u_i|\theta)$ defined in equation 3.

$\theta$ represents all parameters of the model, including any learned representations for the elements of the sets $\mathcal{U}$ and $\mathcal{V}$. An embedding $u_i$ for element $i$ may either be a vector of free parameters, as would be in a fixed vocabulary embedding model, or may be the output of a model on a raw input: $u_i = f_u(\phi_u(i))$, as would be in a twin network model. In addition, each half of the pair $(u_i, v_j)$ may represent a different input space, as in the DSSM model.

## 3.2 BINARY CROSS ENTROPY LOSS

We may simply model the relevancy $y_{ij}$ for the pair $(u_i, v_j)$ with a binary cross entropy loss:

$$l(y_{ij}, i, j | \theta) = - y_{ij} \log\left(\hat{p}(y_{ij} | i, j, \theta)\right) - \beta(1 - y_{ij}) \log\left(1 - \hat{p}(y_{ij} | i, j, \theta)\right). \tag{5}$$

where $\hat{p}$ is the learned estimate for $E[y_{ij}|i, j, \theta]$, and $\beta$ is a scalar hyperparameter for tuning the relative importance of positive and negative samples. One standard choice for $\hat{p}$ in representation learning is to take

$$\hat{p}(y_{ij}|i, j, \theta) = s_\sigma(u_i, v_j) := \sigma\left(\alpha \frac{u_i^T v_j}{||u_i||||v_j||}\right), \tag{6}$$

where $\sigma$ is the logistic function and $\alpha$ is a scalar. As the logistic function saturates quickly, the embeddings $u_i$ and $v_j$ do not need to be extremely close (when $y_{ij}$ is positive) in order to achieve low error. Thus, to encourage representations that are amenable to hashing retrieval, we might consider other transformations of the embeddings that do not saturate so quickly. For example, one may take a polynomial transformation of cosine similarity:

$$\hat{p}(y_{ij}|i, j, \theta) = s_c(u_i, v_j)^K := \frac{1}{2^K}\left(1 + \frac{u_i^T v_j}{||u_i||||v_j||}\right)^K, \tag{7}$$

or a polynomial transformation of the angular similarity:

$$\hat{p}(y_{ij}|i, j, \theta) = \psi(u_i, v_j)^K = \left(1 - \frac{1}{\pi}\cos^{-1}\left(\frac{u_i^T v_j}{||u_i||||v_j||}\right)\right)^K. \tag{8}$$

The $\hat{p}(y_{ij}|i, j, \theta) = \psi(u_i, v_j)^K$ choice has a natural interpretation of using the SimHash collision probability under a $K$ bit hash as the estimation function. Intuitively, we are training representations whose collision probability distribution under SimHash has minimum cross entropy with the pairwise label distribution $y$. Embeddings trained with equation 8 are termed Locality Sensitive Embeddings (LSE) and are the proposed method of this paper. Deterministic thresholding is still used to derive binary embeddings from dense versions.

DCH Cao et al. (2018) introduced the following similarity measure for defining the loss:

$$\hat{p}(y_{ij}|i, j, \theta) = s_h(u_i, v_j) := \frac{\gamma}{\gamma + \frac{d}{2}\left(1 - \frac{u_i^T v_j}{||u_i||||v_j||}\right)}. \tag{9}$$

Table 1: Results of Tuned Models on SBM Experiment

| Model | $K$ | $\beta$ | Hard Negatives? | $\lambda$ | Prec | Rec | F1 |
|-------|-----|---------|-----------------|-----------|------|-----|-----|
| COS | 4 | 1 | False | 0.3 | $0.724 \pm 0.030$ | $0.679 \pm 0.028$ | $0.701 \pm 0.027$ |
| DCH | 2 | 2 | False | 3.0 | $0.924 \pm 0.021$ | $0.916 \pm 0.008$ | $0.920 \pm 0.012$ |
| DHN | 2 | 4 | False | 0.3 | $0.415 \pm 0.017$ | $0.124 \pm 0.010$ | $0.191 \pm 0.013$ |
| LSE | 2 | 1 | True | 0.1 | $0.992 \pm 0.005$ | $0.986 \pm 0.005$ | $0.989 \pm 0.005$ |

### 3.3 TOPOLOGICAL ANALYSIS

The SimHash method and the angular similarity can be used for studying the topologies induced by the different similarity measures in the previous section.

**Theorem 1.** *Let $B(q_i) = N(\delta, q_i, 1-s)$ denote a ball around $q_i$ with radius $\delta$ under the $1-s$ distance. For an arbitrary point $q_j \in B(q_i)$, we can consider the probability $q_j$ and $q_i$ will collide under SimHash – denote this with $P_s(\delta)$. Then,*

1. *(LSE) $P_\psi(\delta) \geq 1 - \delta$*

2. *(COS) $P_{s_c}(\delta) \geq 1 - \frac{2\sqrt{\delta}}{\pi} - O(\delta^{\frac{3}{2}})$*

3. *(DCH) $P_{s_h}(\delta) \geq 1 - \left(\frac{4\gamma\delta}{\pi^2 d(1-\delta)}\right)^{\frac{1}{2}} - O(\delta^{\frac{3}{2}})$*

Proof in Appendix. These bounds are tight as we know the asymptotic error for each (for LSE there is no error term). Theorem 1 reveals that, for a similarity model trained to tolerance level $\delta$ for positive pairs, under a SimHash algorithm LSE would have linear scaling of a single bit collision probability, while COS and DCH would have sublinear scaling.

Note that for the logistic based similarity, $P_{s_\sigma}(\delta)$ is only well defined for $\alpha > |\log(\delta) - \log(1-\delta)|$ (otherwise $1 - s_\sigma$ cannot be below $\delta$.) Any analysis here requires choosing a rate for $\alpha$.

There is also a relationship between Angular and Hamming distances. Angular distance can be viewed as a dimension scaled version of Hamming distance applied to randomly rotated inputs.

**Lemma 1.** *Let $b(q_i)$ be the vector indicating signs of $q_i$, that is $b(q_i)_m := \mathbb{1}[q_{im} > 0]$. Denote the Hamming distance of the sign vectors as $\rho_H(q_i, q_j) := ||b(q_i) - b(q_j)||_1$ which defines a semimetric on $\mathbb{R}^d$. Take $R$ as a uniformly random orthogonal matrix. Then*

$$1 - \psi(q_i, q_j) = \frac{1}{d}\mathbb{E}_R[\rho_H(Rq_i, Rq_j)]. \tag{10}$$

Proof in Appendix. Lemma 1 demonstrates that angular distance may be viewed as an expectation of a dimension-scaled Hamming distance, where the expectation is taken with respect to the choice of basis. In other words, minimizing angular distance is equivalent to minimizing the Hamming semimetric $\rho_H$ averaged over all possible bases.

## 4 EXPERIMENTS

In this section, we compare representations trained using equation 8 (LSE), using equation 7 (COS), using DHN's logistic based similarity, and using DCH. All methods use the quantization loss in equation 3 from Zhu et al. (2016), except DCH which uses the quantization loss from Cao et al. (2018).

### 4.1 SYNTHETIC DATA

We generate data from a Stochastic Block Model (SBM) (Holland et al. (1983)) with 500 factions, 10 individuals per faction, and the probability of an edge appearing between two individuals belonging to factions $i$ and $j$ governed by the matrix of probabilities $W \in \mathbb{R}^{500 \times 500}$ with $W_{ij} = 0.8\mathbb{I}(|i-j| = 0) + 0.1\mathbb{I}(0 < |i-j| < 3) + \epsilon\mathbb{I}(|i-j| >= 3)$. The resulting cooccurrence matrix $D \in \{0, 1\}^{5000 \times 5000}$

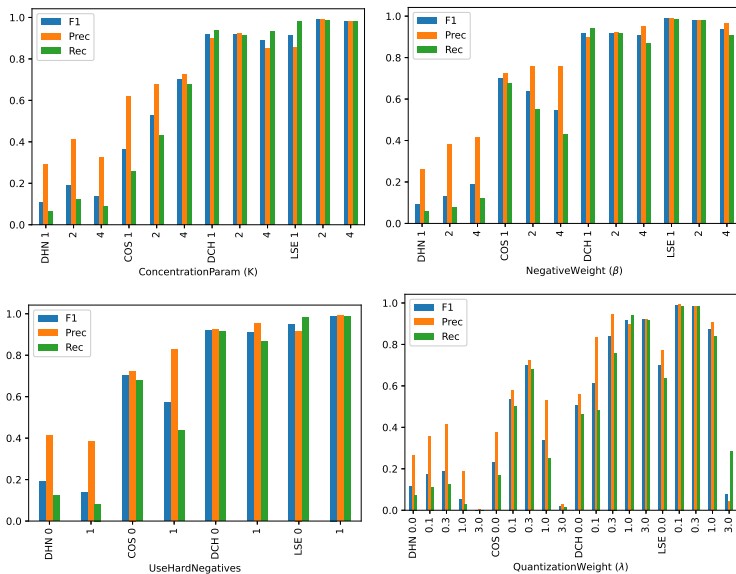

Figure 2: Results of the SBM experiment hyperparameter search, sliced by different variables. Each set of bars represents the scores of the highest F1 model with the listed constraint.

Table 2: Mean Average Precision on Image Models

| Datasets | Cifar-10 | | | | NUS Wide | | | | ImageNet | | | |
|---|---|---|---|---|---|---|---|---|---|---|---|---|
| Methods\Metrics | 64 | 48 | 32 | 16 | 64 | 48 | 32 | 16 | 64 | 48 | 32 | 16 |
| DHN | 0.779 | 0.776 | 0.773 | 0.777 | 0.821 | 0.818 | 0.809 | 0.790 | 0.518 | 0.473 | 0.423 | 0.400 |
| DCH | 0.774 | 0.777 | 0.780 | 0.772 | 0.763 | 0.761 | 0.759 | 0.742 | 0.564 | 0.583 | 0.607 | 0.558 |
| COS | 0.802 | 0.802 | 0.797 | 0.778 | 0.792 | 0.788 | 0.775 | 0.757 | 0.638 | 0.634 | 0.616 | 0.558 |
| LSE | 0.789 | 0.786 | 0.788 | 0.772 | 0.758 | 0.753 | 0.742 | 0.732 | 0.594 | 0.586 | 0.567 | 0.505 |

is roughly block diagonal, with additional edges appearing from "nearby" factions at a lower rate. The task is to hash the individuals from each faction together, while separating them from all other factions. In order to enable higher precision of the resulting models, a set of hard negatives is generated by taking pairs $(i, j)$ such that $(D^T D)_{ij} > 0$ and $D_{ij} = 0$. Easy negatives are also generated by taking pairs at random and assuming a cooccurrence of 0, as is common in Noise Contrastive Estimation (Gutmann & Hyvärinen (2010)) inspired methods.

Each individual is given a free parameter $u_i$ which is the output of a embedding layer followed by `tanh` activation. This output is dropped-out (shared randomization for each half of the training pair) before the cosine similarity computation. Batches are constructed from 1024 positive pairs, and 3072 negative pairs (1024 of which may either be hard or easy negatives, determined as a hyperparameter.)

We trained all models with 32 dimensional representations, for 50 epochs, where 1 epoch is the number of batches required to iterate through all positive pairs. We explore 4 hyperparameters, $K$, $\beta$, $\lambda$, and the use of hard negative samples. During evaluation, we retrieve all individuals which have the same binarized embedding as the query individual. We measure precision, recall and F1-score with the data generating factions as the target. 10 trials are repeated for each hyperparameter setting, and the mean over trials is reported. Figure 2 shows a detailed view of the hyperparameter tuning, and Table 1 shows the chosen hyperparameters for each model when ranking by F1-score. DCH and LSE are competitive, however the LSE model is able to achieve surprisingly accurate recovery of the data generating structure with an F1 of nearly 0.99.

Table 3: Tuned Parameters and Performance of 100-Epoch Models on OSCAR dataset

| Model | $K$ | $\beta$ | $\lambda$ | Tr. Prec. | Tr. Rec | Tr. F1 | WP | Ts. HR@50 | Ts. HR@500 | Clusters |
|-------|-----|---------|-----------|-----------|---------|--------|-------|-----------|------------|----------|
| DHN | 12 | 1 | 0.3 | 0.031 | 0.005 | 0.008 | 0.042 | 0.004 | 0.009 | 14K |
| COS | 16 | 2 | 0.3 | 0.301 | 0.186 | 0.230 | 0.279 | 0.040 | 0.100 | 298K |
| DCH | 4 | 4 | 3.0 | 0.345 | 0.185 | 0.241 | 0.264 | 0.045 | 0.115 | 253K |
| LSE | 8 | 4 | 0.3 | 0.412 | 0.260 | 0.319 | 0.291 | 0.048 | 0.150 | 313K |

Table 4: Example Clusters Discovered

| Cluster Words | Cluster Size | Author Annotations |
|---------------|--------------|--------------------|
| basil, honey, vinegary, ribeye, grilling, ... | 1134 | Generic Food |
| vegan, veganism, cleaneating, vegano, plantpower, ... | 28 | Vegan Terms |
| scleractinia, pertusa, pistillata, acropora, hystrix, ... | 14 | Coral Species |
| bristle, tinged, mohair, metallic, inseam, ... | 1208 | Generic Style/Fashion |

## 4.2 Hamming Ball Image Retrieval on Image Datasets

The experiment on image datasets is motivated from state of the art methods which build binary embeddings directly and use hashing retrieval for image similarity search. In this section we demonstrate that LSE is appropriate for these tasks as well. We follow the experimental setup as in Zhu et al. (2016) on three datasets: Cifar-10 Krizhevsky et al. (2009), NUS-WIDE Chua et al. (2009) and ImageNet Russakovsky et al. (2015). Cifar-10 is a dataset consisting of 10 categories and 60000 color images of size $32 \times 32$. We use the same data splits (available online) as Zhu et al. (2016): 500 images per category in training set, 100 images per category it test query set and the remaining 54000 images are used as database. We again follow Zhu et al. (2016) for experiments with NUS-WIDE dataset: 149736 images that are associated with 21 most frequent categories as database and 2100 images as queries, and 10500 images from the database as training set. For ImageNet, we use the same 100 categories as Cao et al. (2017) as the indexing database and the 13000/5000 image train/test split. We follow Cao et al. (2018) and use pretrained AlexNet as described. We take the open-source code for deep hashing methods [1] (Zhu et al. (2016)) and add the LSE model. Results are shown in Table 2. We show results for binary embedding size of 64 bits, 48 bits, 32 bits, and 16 bits. We do not alter any other setting and use $K = 1$, $\lambda = 0.1$ for all models. In pilot experiments we did not find any significant improvement for higher $K$ for any of the methods. All baseline papers use mean average precision for the evaluation (Cao et al. (2018); Zhu et al. (2016)) which is the evaluation method we adopt for this experiment. The LSE model is comparable to the baseline methods. Cifar10 results are statistically significant with the p-value of $7.5 \times 10^{-4}$, NUS Wide with the p-value of $2.81 \times 10^{-146}$ and Imagenet with the p-value of $7.99 \times 10^{-4}$ according to widely used Iman Daveport test (Garcia & Herrera, 2008). We want to point out that all the experimental results (originally in the respective baseline papers and in this paper) on DCH and DHN have used pretrained AlexNet. The usage of pretrained network and well separated categories reduces the need for a model with a strong inductive bias like LSE. Nonetheless LSE remains a good choice for these type of tasks as well.

## 4.3 OSCAR Cooccurrence Model

Finally, we compare all methods on a cooccurrence matrix generated from the OSCAR English dataset (Ortiz Suarez et al. (2019); Ortiz Suarez et al. (2020)). We take the deduplicated version of the corpus (1.2TB compressed) and generate an initial symmetric cooccurrence matrix by counting word pairs with a window of size 10, inversely weighting the counts by the distance of the words within the sentence, as in Pennington et al. (2014). This initial matrix is then filtered to remove extremely common terms, extremely rare terms. Additional filtering based on row and column normalized cooccurrences is used to retain pairs that are atypical compared to the marginal frequencies of the two terms. For each row, the top 100 pairs ranked by the original cooccurrence are kept, and the resulting binary matrix is symmetrized. The resulting cooccurrence matrix $D$ has 660K unique

---

[1]`https://github.com/swuxyj/DeepHash-pytorch`

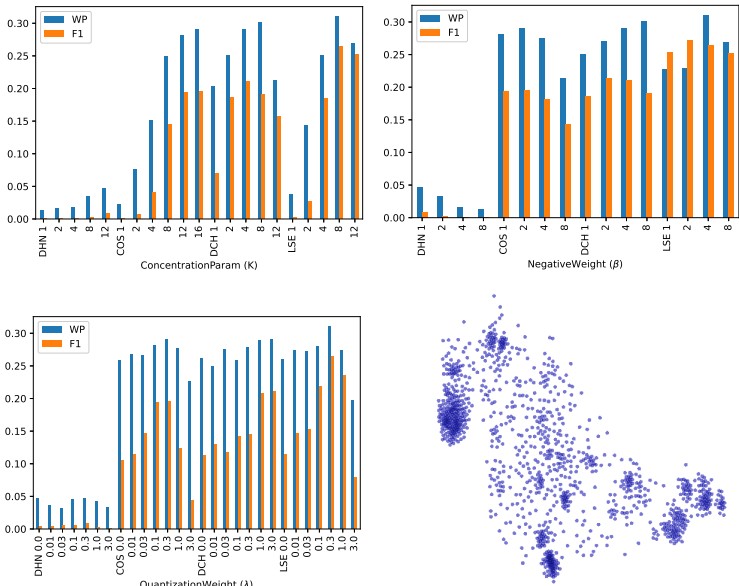

Figure 3: Results of the OSCAR experiment hyperparameter search, sliced by different variables. Each set of bars represents the scores of the highest combined WP+F1 measures (where WP is Wu-Palmer similarity – described in main text) with the listed hyperparameter value fixed. F1 score are computed on a query set randomly sampled from the 660K vocabulary, and is computed by performing retrieval on the full 660K vocabulary. Wu-Palmer similarity is computed on the 46K WordNet evaluation vocabulary, with a query set sampled from the WN vocabulary. (**bottom right**) T-SNE plot of the dense embeddings assigned to a single hash cluster ("Generic Food"), revealing additional structure

words, with 16M nonzero entries. A set of hard negatives is generated by taking pairs $(i, j)$ such that $(D^T D)_{ij} > 0$ and $D_{ij} = 0$. 4600 pairs from $D$ are held out for the final evaluation.

Popular models for text data frequently allow for each word's representation to have multiple contexts, as in topic modelling (Hofmann (1999); Blei et al. (2003)) or multi-sense embedding models (Nguyen et al. (2017)). To incorporate multiple context representations into semantic hashing methods, we represent each word $i$ with $L$ embeddings $u_{il}$ (these are free parameters per word and no subword information is used.) The maximum among all pairwise cosine similarities is then taken as the base similarity function:

$$s(u_i, u_j) = \max_{l,m} \frac{u_{il}^T u_{jm}}{||u_{il}||||u_{jm}||}. \tag{11}$$

This base similarity is then used in place of cosine similarity in defining the loss for all models[2]. At retrieval time, a query word is mapped to its $L$ hashes, corresponding to $L$ "clusters." The union of the $L$ clusters is the retrieved set for the query – no search is performed and the only data structures used are hash tables. As each word is associated with $L$ hashes, this model may be understood as a "word2hashes" method.

The base architecture used for all models is a 32 dimension `tanh` activated embedding layer followed by dropout (with shared randomization across all $2L$ embeddings,) with $L = 3$. Following the dropout layer is equation 11. Batches are constructed from 8192 positives, 8192 hard negatives, and 16384 easy negatives. Models are trained for 20 epochs through the positive set using 2 GPUs.

We utilize a semantic quality measure based on Wu-Palmer similarity (WP) (Wu & Palmer (1994)) on WordNet (WN) (Miller (1995); Fellbaum et al. (1998)). We take all nouns, verbs and adjectives from the WordNet corpus and remove all words with no hypernyms (these are typically isolated nodes in the WordNet graph for which WP values are not available.) The intersection with the 660K

---

[2]As DHN uses the unnormalized inner product, for which the max operation has undesirable properties, we modify the DHN implementation to use $5s(u_i, u_j)$ as the input to the logistic function.

vocabulary leaves 46K words, which we index based on the semantic hashing models. For each query word $w$ and its retrieved set $V(w)$, the average WP similarity is computed across all pairs $w, v$ with $v \in V(w)$. Self-pairs are removed, and empty $V(w)$ are given 0 values. This WP measure is bounded between 0 and 1, with higher values indicating more semantically meaningful clusters.

Figure 3 shows WP of the models on a 1K word query set (taken randomly from the 46K WN vocabulary) which we use as a tuning data set. We also report F1 score on a 1K query set sampled from the 660K vocabulary to evaluate how well each model reconstructs the training data. All models used the same tuning grid, except COS for which $K = 16$ was added, as the initial sweep showed potentially large improvement by expanding the grid for the COS model.

Table 3 shows the final model comparison. The hyperparameters with highest WP per model are taken and models for each are trained with 100 epochs. We compare the scores of these models on a non-tuning set of 1K queries from WN for WP, and 1K queries from the full vocab for Precision, Recall, and F1. In addition, we evaluate a HitRatio (HR) score on the heldout 4600 pairs, where all colliding words for a query are retrieved, and if the target word appears in the top $n$ items ranked by cosine similarity (of the dense embeddings,) the query achieves a HR of 1. This is the only measure to use the dense embeddings. We also report the number of non-singleton clusters. As can be seen, LSE outperforms the baselines on training, test, and semantic quality measures.[3]

We display some example queries and their retrieved hash siblings from the 100 epoch LSE model in Table 7. T-SNE Maaten & Hinton (2008) plots of the dense embeddings on WordNet vocabulary are shown in Figures 5, 6, and 7. Within the discrete hash based clusters used in retrieval, there is still additional structure in the dense embeddings that may be leveraged. This can be seen from T-SNE plots (Figures 8 and 9) of the dense embeddings that collide together in the "Generic Food" cluster seen in Table 7.

## 5    CONCLUSION

We extend semantic hashing methods to problems with substantial label noise and to the exact hashing retrieval case via the introduction of Locality Sensitive Embeddings, which leverage angular similarity as the main component of an output prediction. The learned representations show superior performance in the exact hashing retrieval setting. We applied LSE to a multiple-context representation learning model to a cooccurrence matrix generated from the OSCAR English corpus, producing a "word2hashes" model which is novel to the best of the authors' knowledge.

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

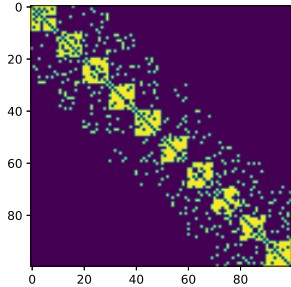 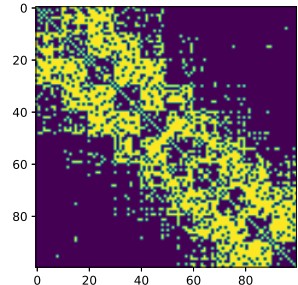

Figure 4: **(left)** Subset of SBM dataset, indicating near block-diagonal structure. **(right)** Hard negatives sampled for SBM dataset

Olga Russakovsky, Jia Deng, Hao Su, Jonathan Krause, Sanjeev Satheesh, Sean Ma, Zhiheng Huang, Andrej Karpathy, Aditya Khosla, Michael Bernstein, et al. Imagenet large scale visual recognition challenge. *International journal of computer vision*, 115(3):211–252, 2015.

Ruslan Salakhutdinov and Geoffrey Hinton. Semantic hashing. *International Journal of Approximate Reasoning*, 50(7):969–978, 2009.

Atsushi Sato and Keiji Yamada. Generalized learning vector quantization. In *Advances in neural information processing systems*, pp. 423–429, 1996.

Kilian Q Weinberger, John Blitzer, and Lawrence K Saul. Distance metric learning for large margin nearest neighbor classification. In *Advances in neural information processing systems*, pp. 1473–1480, 2006.

Zhibiao Wu and Martha Palmer. Verb semantics and lexical selection. *arXiv preprint cmp-lg/9406033*, 1994.

Eric P Xing, Michael I Jordan, Stuart J Russell, and Andrew Y Ng. Distance metric learning with application to clustering with side-information. In *Advances in neural information processing systems*, pp. 521–528, 2003.

Han Zhu, Mingsheng Long, Jianmin Wang, and Yue Cao. Deep hashing network for efficient similarity retrieval. In *Thirtieth AAAI Conference on Artificial Intelligence*, 2016.

## A  APPENDIX

### A.1  EFFECT OF HARD NEGATIVES AND ABLATION STUDY

In this section we study the impact of hard negatives. Recall from the main text that we define hard negatives from a dataset $D$ by taking pairs where $(D^T D)_{ij} > 0$ and $D_{ij} = 0$. See Figure 4 for a diagram showing these hard negative pairs on the SBM synthetic data.

We performed an experiment on the OSCAR dataset in which a modified loss function is used where hard negatives samples are weighted (that is, easy negatives are always given weight of 1.) These tuned models are compared to the tuned models of the main text in Table 5. This tuning gives a modest improvement to COS and DCH models in terms of WP (and a degradation for LSE,) while keeping F1 much the same. However, DCH is not able to obtain the same F1 measure as LSE (in either tuning.) In addition, the LSE model from the original tuning outperforms the other models in WP+F1. Also note that the DHN model improves substantially on the removal of Hard Negatives, however it still remains the worst performing algorithm of the four.

We also use this experiment to perform an ablation study, see Table 6. DHN is most competitive when hard negatives are removed, achieving the highest WP (but still lowest F1.) It is also this case where we see LSE achieve a Recall of 0.54 – this metric is essentially traded-off for Precision

Table 5: 20-Epoch Models on OSCAR Dataset with Various Tuning Criteria

| Tuning Criteria | Model | $K$ | $\beta$ | $\lambda$ | Tr. Prec. | Tr. Rec | Tr. F1 | WP |
|---|---|---|---|---|---|---|---|---|
| | DHN | 12 | 1 | 0.0 | 0.018 | 0.003 | 0.004 | 0.047 |
| Max WP | COS | 16 | 2 | 0.3 | 0.269 | 0.154 | 0.196 | 0.291 |
| | DCH | 8 | 8 | 1.0 | 0.141 | 0.174 | 0.156 | 0.304 |
| | LSE | 8 | 4 | 0.3 | 0.409 | 0.196 | 0.265 | 0.310 |
| | DHN | 12 | 1 | 0.3 | 0.027 | 0.004 | 0.008 | 0.046 |
| Max F1 | COS | 16 | 1 | 0.3 | 0.218 | 0.195 | 0.206 | 0.260 |
| | DCH | 4 | 2 | 3.0 | 0.259 | 0.198 | 0.225 | 0.250 |
| | LSE | 8 | 2 | 0.3 | 0.295 | 0.250 | 0.271 | 0.230 |
| | DHN | 12 | 1 | 0.3 | 0.027 | 0.004 | 0.008 | 0.046 |
| Max WP+F1 | COS | 16 | 2 | 0.3 | 0.269 | 0.154 | 0.196 | 0.291 |
| | DCH | 4 | 4 | 3.0 | 0.335 | 0.153 | 0.210 | 0.290 |
| | LSE | 8 | 4 | 0.3 | 0.409 | 0.196 | 0.265 | 0.310 |
| | DHN | 12 | 0 | 0.03 | 0.109 | 0.088 | 0.098 | 0.241 |
| (Hard Negative $\beta$) Max WP+F1 | COS | 12 | 1 | 0.3 | 0.279 | 0.153 | 0.197 | 0.306 |
| | DCH | 4 | 2 | 1.0 | 0.238 | 0.197 | 0.215 | 0.305 |
| | LSE | 8 | 2 | 0.3 | 0.272 | 0.260 | 0.266 | 0.285 |

Table 6: Ablation Study of 20-Epoch Models on OSCAR Dataset (Hard Negative $\beta$ formulation)

| Ablated Paramater | Model | $K$ | $\beta$ | $\lambda$ | Tr. Prec. | Tr. Rec | Tr. F1 | WP |
|---|---|---|---|---|---|---|---|---|
| | DHN | 12 | 0 | 0.03 | 0.109 | 0.089 | 0.098 | 0.241 |
| Hard Negative Weight $\beta$ | COS | 8 | 0 | 0.10 | 0.137 | 0.263 | 0.178 | 0.162 |
| | DCH | 2 | 0 | 3.00 | 0.156 | 0.366 | 0.219 | 0.076 |
| | LSE | 8 | 0 | 0.30 | 0.139 | 0.541 | 0.221 | 0.078 |
| | DHN | 12 | 0.00 | 0 | 0.098 | 0.062 | 0.076 | 0.218 |
| Quantization Weight $\lambda$ | COS | 8 | 0.25 | 0 | 0.156 | 0.086 | 0.111 | 0.264 |
| | DCH | 4 | 4.00 | 0 | 0.224 | 0.075 | 0.112 | 0.267 |
| | LSE | 12 | 8.00 | 0 | 0.163 | 0.091 | 0.117 | 0.275 |
| | DHN | 1 | 0.0 | 0.03 | 0.004 | 0.001 | 0.001 | 0.107 |
| Concentration $K$ | COS | 1 | 0.0 | 0.00 | 0.006 | 0.005 | 0.005 | 0.141 |
| | DCH | 1 | 0.5 | 1.00 | 0.121 | 0.061 | 0.082 | 0.218 |
| | LSE | 1 | 0.0 | 0.03 | 0.005 | 0.021 | 0.008 | 0.097 |

and WP when increasing the negative weight. All methods have comparable performance when the quantization loss is removed. And finally, DCH performs the best when $K = 1$.

### A.2 PROOFS

**Theorem 1.** *Let $B(q_i) = N(\delta, q_i, 1 - s)$ denote a ball around $q_i$ with radius $\delta$ under the $1 - s$ distance. For an arbitrary point $q_j \in B(q_i)$, we can consider the probability $q_j$ and $q_i$ will collide under SimHash – denote this with $P_s(\delta)$. Then,*

1. *(LSE) $P_\psi(\delta) \geq 1 - \delta$*

2. *(COS) $P_{s_c}(\delta) \geq 1 - \frac{2\sqrt{\delta}}{\pi} - O(\delta^{\frac{3}{2}})$*

3. *(DCH) $P_{s_h}(\delta) \geq 1 - \left( \frac{4\gamma\delta}{\pi^2 d(1-\delta)} \right)^{\frac{1}{2}} - O(\delta^{\frac{3}{2}})$*

*Proof.* (1): $1 - \psi(q_i, q_j) \leq \delta$ by definition, so $P_\psi(\delta) \geq 1 - \delta$.

For each of the following, we will use the expansion via the Frobenius method: $\cos^{-1}(1 - \delta) = \sqrt{2\delta} + O(\delta^{\frac{3}{2}})$.

(2): $1 - s_c(q_i, q_j) \leq \delta$. Substituting $\frac{q_i^T q_j}{||q_i||||q_j||} = \cos(\pi(1 - \psi(q_i, q_j)))$ in $s_c$ and rearranging gives (note $\cos^{-1}$ is monotonically decreasing on $(0, 1)$)

$$1 - s_c(q_i, q_j) = \frac{1}{2} \left( 1 - \frac{q_i^T q_j}{||q_i||||q_j||} \right) \leq \delta$$

$$\cos(\pi(1 - \psi(q_i, q_j))) \geq 1 - 2\delta$$

$$1 - \psi(q_i, q_j) \leq \frac{1}{\pi} \cos^{-1}(1 - 2\delta) = \frac{2\sqrt{\delta}}{\pi} + O(\delta^{\frac{3}{2}}).$$

(3): $1 - s_h(q_i, q_j) \leq \delta$. Proceeding as above

$$1 - s_h(q_i, q_j) = 1 - \frac{\gamma}{\gamma + \frac{d}{2} \left( 1 - \frac{q_i^T q_j}{||q_i||||q_j||} \right)} \leq \delta$$

$$\cos(\pi(1 - \psi(q_i, q_j))) \geq 1 - \frac{2\gamma\delta}{d(1 - \delta)}$$

$$1 - \psi(q_i, q_j) \leq \frac{1}{\pi} \cos^{-1} \left( 1 - \frac{2\gamma\delta}{d(1 - \delta)} \right) = \left( \frac{4\gamma\delta}{\pi^2 d(1 - \delta)} \right)^{\frac{1}{2}} + O(\delta^{\frac{3}{2}}).$$

$\square$

Note that for the logistic based similarity, $P_{s_\sigma}(\delta)$ is only well defined for $\alpha > |\log(\delta) - \log(1 - \delta)|$ (otherwise $1 - s_\sigma$ cannot be below $\delta$.) Any analysis here requires choosing a rate for $\alpha$.

**Lemma 1.** *Let $b(q_i)$ be the vector indicating signs of $q_i$, that is $b(q_i)_m := \mathbb{1}[q_{im} > 0]$. Denote the Hamming distance of the sign vectors as $\rho_H(q_i, q_j) := ||b(q_i) - b(q_j)||_1$ which defines a semimetric on $\mathbb{R}^d$. Take $R$ as a uniformly random orthogonal matrix. Then*

$$1 - \psi(q_i, q_j) = \frac{1}{d} \mathbb{E}_R [\rho_H(Rq_i, Rq_j)]. \tag{12}$$

*Proof.* Consider a modified SimHash algorithm where $z'$ is taken uniformly at random from the standard basis vectors $\{e_m\}_{m \in [1,...d]}$ and let $\bar{h}(q_i) := \mathbb{1}[q_i^T z' > 0]$ denote this hash. The collision probability is simply the chance that two given embeddings share the same sign for a randomly chosen dimension, so

$$Pr[\bar{h}(q_i) = \bar{h}(q_j)] = \mathbb{E}_{m \sim (1,...,d)} [\mathbb{1}[q_{im} = q_{jm}]] = 1 - \frac{\rho_H(q_i, q_j)}{d}. \tag{13}$$

Let $z_0 \in \mathbb{R}^d$ be a vector with independent standard Normal entries, and take $z = \frac{z_0}{||z_0||}$, which is equal in distribution to the uniform distribution on the unit sphere. Thus $Rz' \stackrel{d}{=} z$. The original SimHash function $h(q_i) = \mathbb{1}[q_i^T z_0 > 0] = \mathbb{1}[q_i^T z > 0]$, and so $h(q_i) \stackrel{d}{=} \bar{h}(Rq_i)$, and thus

$$\psi(q_i, q_j) = Pr[h(q_i) = h(q_j)] = \mathbb{E}_R \left[ 1 - \frac{\rho_H(Rq_i, Rq_j)}{d} \right]. \tag{14}$$

$\square$

### A.3 EVALUATION METRICS

- Precision - Retrieve all items with the same hash, and compute precision
- Recall - Retrieve all items with the same hash, and compute recall
- F1 - Compute F1-measure using the above Precision and Recall
- WP - Wu-Palmer similarity measure for evaluation of semantic quality of hash groups. Wu-Palmer similarity (Wu & Palmer (1994)) is computed on WordNet (WN) (Miller (1995); Fellbaum et al. (1998)). We take all nouns, verbs and adjectives from the WordNet corpus and remove all words with no hypernyms (these are typically isolated nodes in the WordNet graph for which WP values are not available.) The intersection with the 660K OSCAR vocabulary leaves 46K words, which we index based on the semantic hashing models. For each query word $w$ and its retrieved set $V(w)$, the average WP similarity is computed across all pairs $w, v$ with $v \in V(w)$. Self-pairs are removed, and empty $V(w)$ are given 0 values. This WP measure is bounded between 0 and 1, with higher values indicating more semantically meaningful clusters.
- HR@n - HitRatio score on the heldout (test) 4600 pairs, where all colliding words for a query are retrieved, and if the target word appears in the top $n$ items ranked by cosine similarity (of the dense embeddings,) the query achieves a HR of 1. This is the only measure in the OSCAR experiment to use the dense embeddings.
- Mean Average Precision (MAP) - Consider rank position of each relevant retrieved item (in top $R$) based on Hamming distance - $K_1, K_2, \ldots, K_R$. First calculate precision @ k by setting a rank threshold $k \leq R$ and then ratio of relevant in top k divided by k (ignoring the ranked lower than k). Next step is to calculate average of precision at $0 < r \leq R$. Finally mean average precision is calculated by taking mean of average precision over all queries.
- Recall@2 - Retrieve everything within hamming distance 2 and calculate recall

### A.4 QUALITATIVE PLOTS AND TABLES

Table 7 shows example hash factors retrieved in the OSCAR model for a set of queries. These are constructed from exact hash collisions only – no search or ANN is performed in either the binary representation space or the dense embeddings space. Figures 5, 6 and 7 show TSNE plots for the dense embeddings on the WordNet set. Figures 8 and 9 show TSNE plots for the dense embeddings associated with the single hash that resembles a "Generic Food" cluster. These figures demonstrate there is significant structure remaining in the dense version of the embeddings that is semantically meaningful.

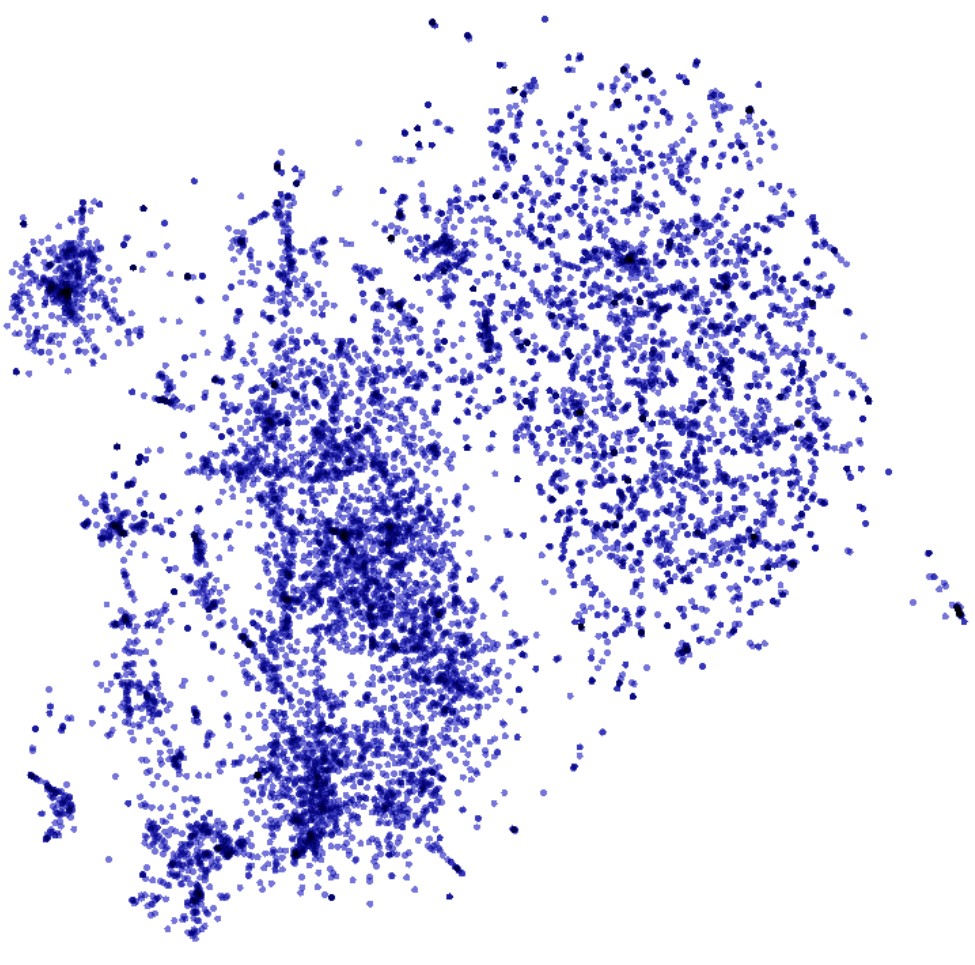

Figure 5: TSNE on the LSE 100 Epoch model on WordNet

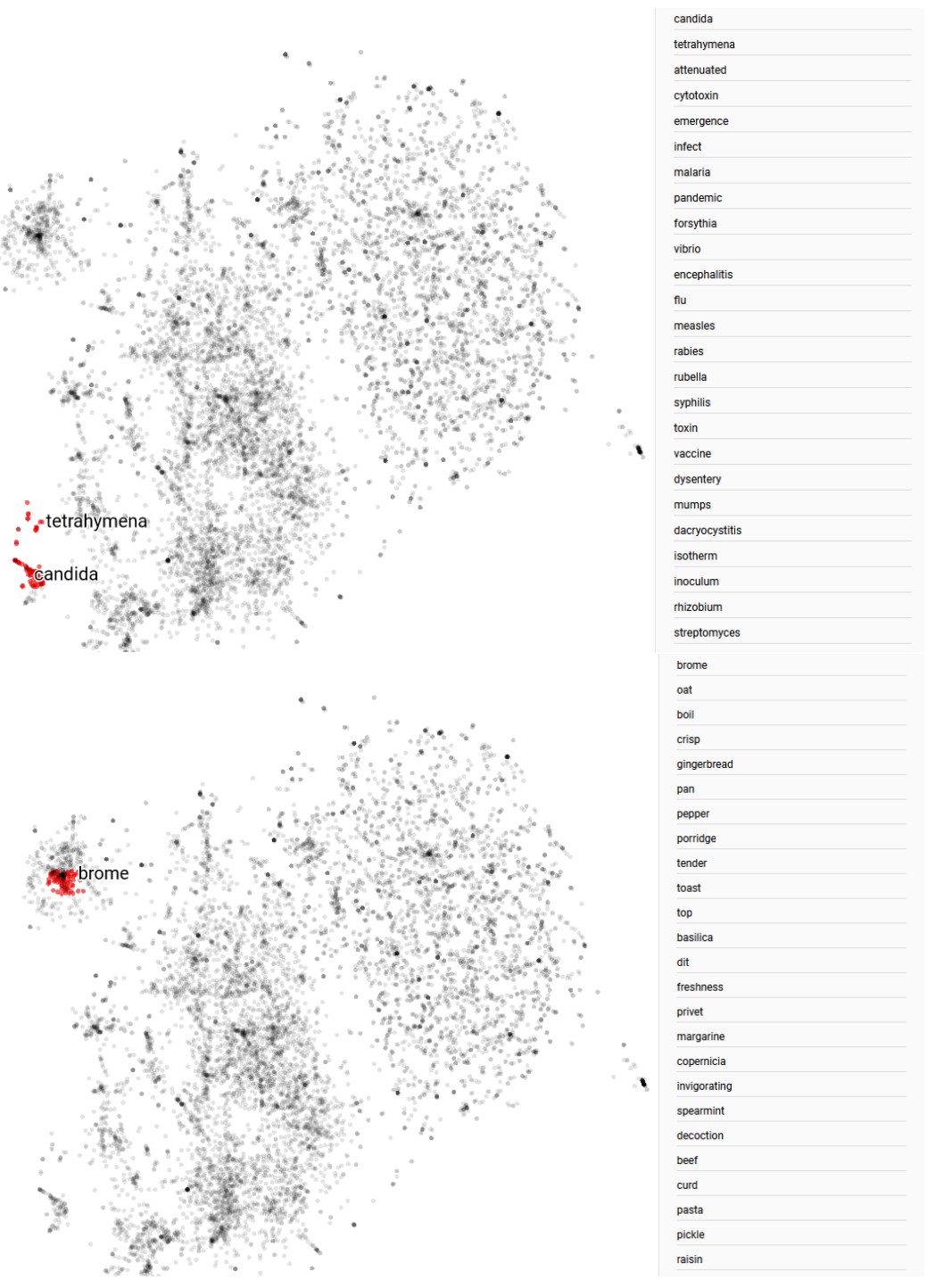

Figure 6: TSNE clusters from the WordNet set

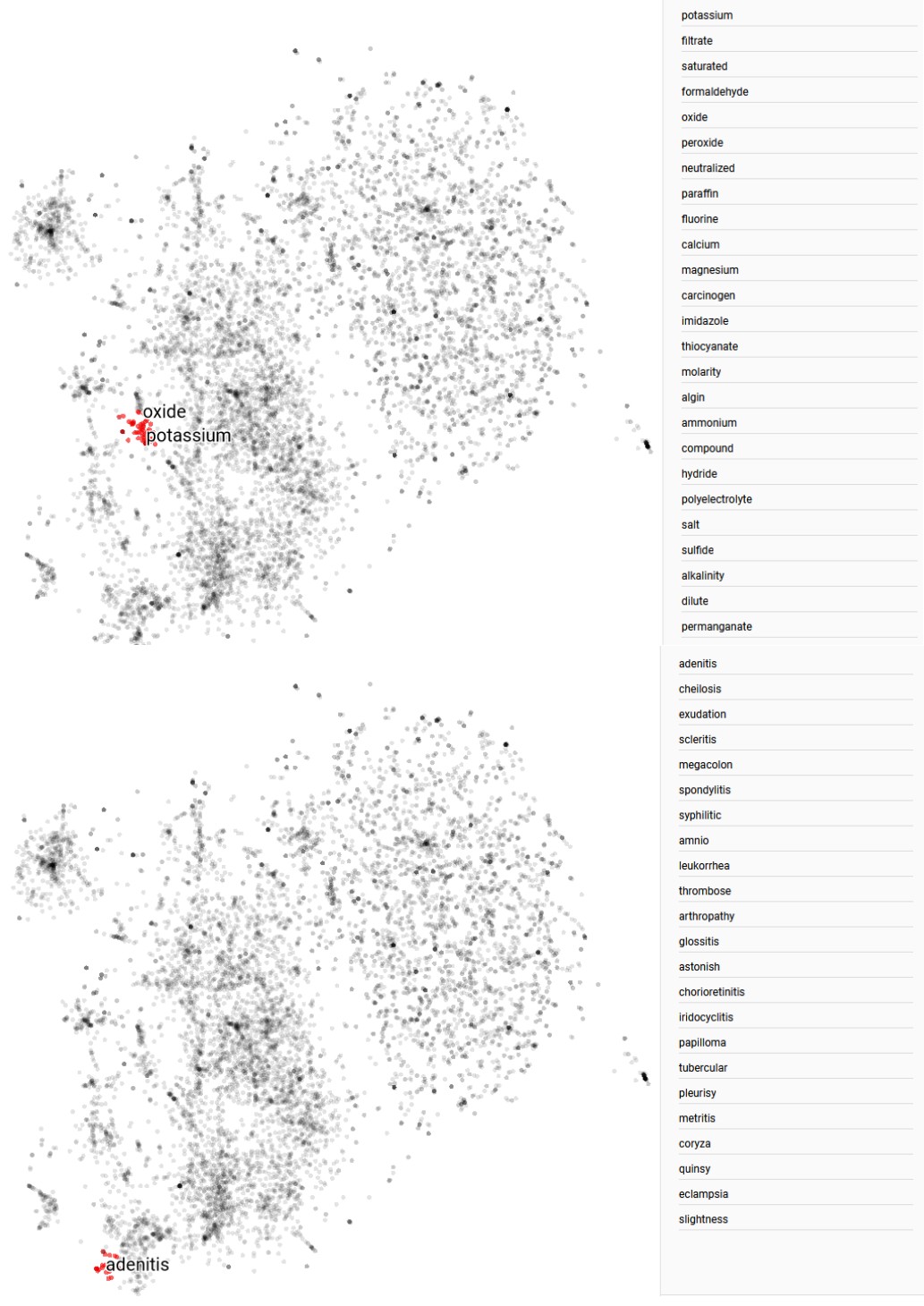

Figure 7: TSNE clusters from the WordNet set

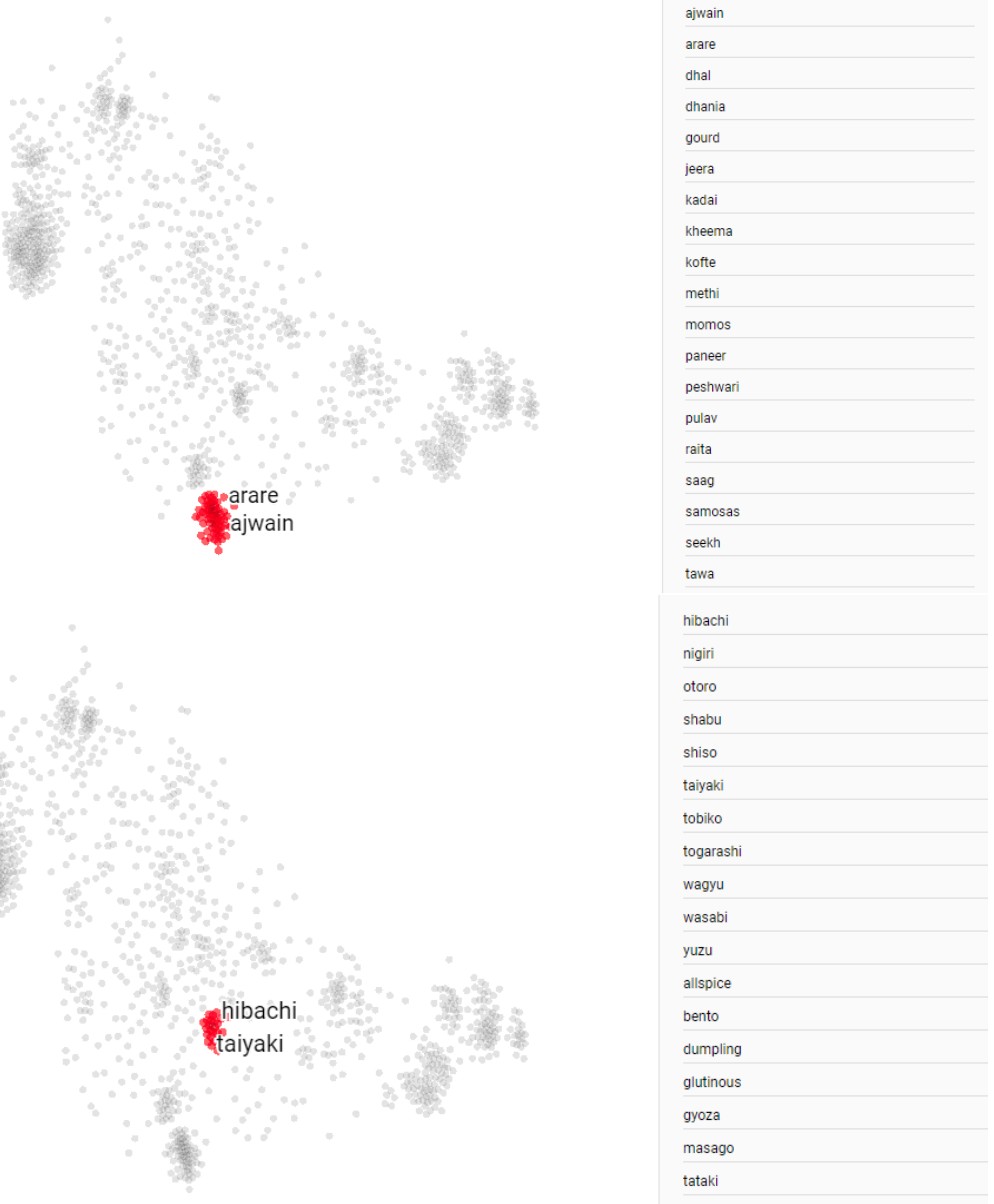

Figure 8: TSNE subclusters from the "Generic Food" Cluster

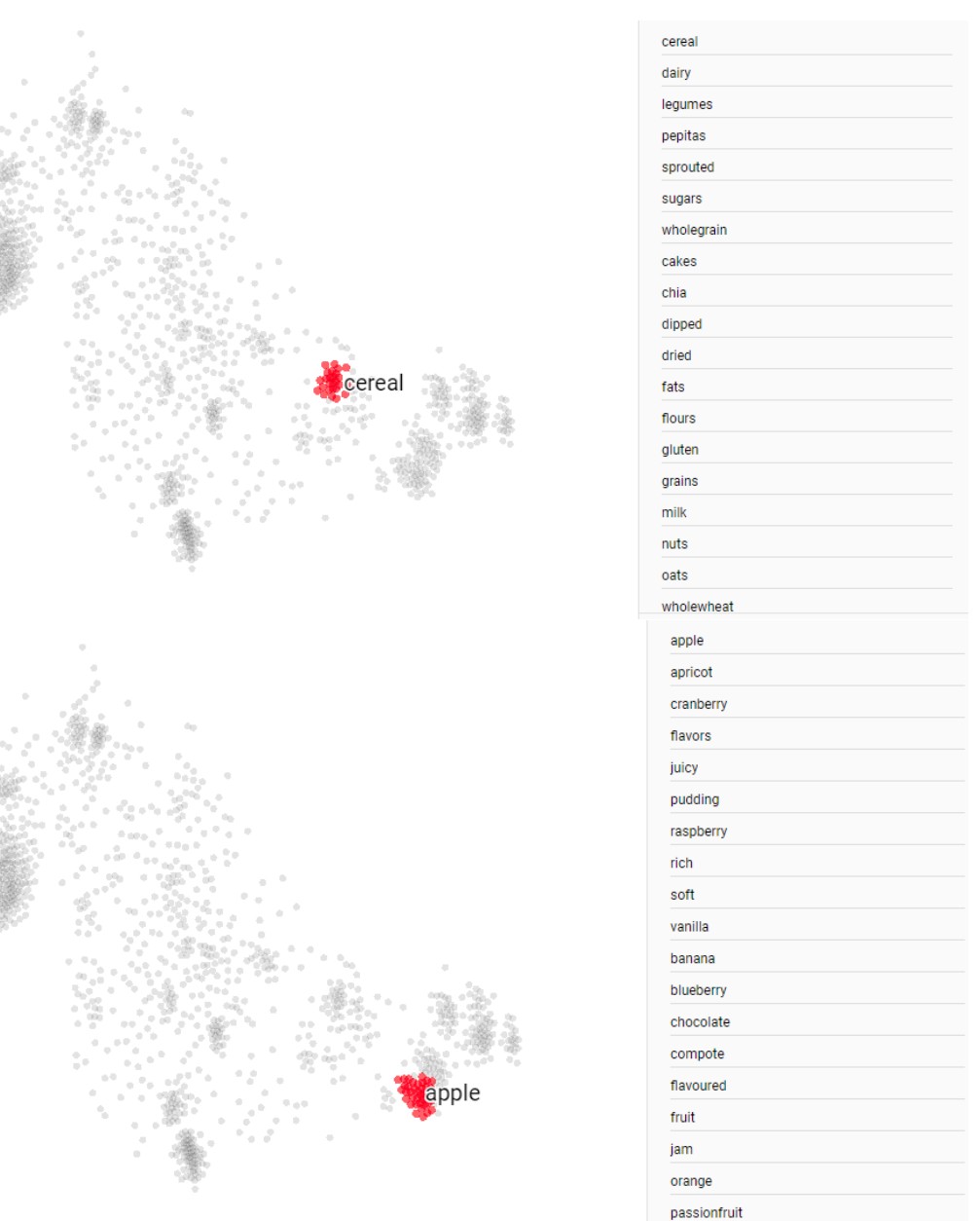

Figure 9: TSNE subclusters from the "Generic Food" Cluster

| Query | Cluster Siblings | Cluster Size | Author Annotations |
|---|---|---|---|
| succulent | stonecrop, sedums, crassula, sempervivum, ... | 10 | Succulent Plants |
| succulent | | 1 | |
| succulent | battered, tossing, tenderized, spareribs, ... | 7 | Enticing Food Terms |
| parser | sax2, lalr, fixpoint, xerces, gson, ... | 22 | Parser Instances |
| parser | fastboot, lwo, eml, compute, osgi, ... | 732 | Generic Technology |
| parser | tokenizer, javacc, tokenizes, jtb, lexer | 6 | Java Tree Builder |
| lazarette | amidship, starboard, portside | 4 | Ship Locations |
| lazarette | cockpit, coaming | 3 | Ship Components |
| lazarette | coatroom, garran, carrels, stowage, locker, ... | 12 | Storage |
| vegan | basil, honey, vinegary, ribeye, grilling, ... | 1134 | Generic Food |
| vegan | vegatarian | 2 | Portmanteau |
| vegan | veganism, cleaneating, vegano, plantpower, ... | 28 | Vegan Terms |
| coral | scleractinia, pertusa, pistillata, acropora, hystrix, ... | 14 | Coral Species |
| coral | yellow, root, crocheted, soles, linen, ... | 1208 | Generic Fashion |
| coral | | 1 | |
| bronchitis | eyedrops, fever, myalgia, sinusitis, fasciculations, ... | 187 | Generic Health |
| bronchitis | | 1 | |
| bronchitis | pneumonia | 2 | Respiratory Infections |
| split | | 1 | |
| split | subsections, empirical, broader, alexa, cafes, ... | 5455 | Generic English |
| split | ubli, imotski, zlatni, milna, dalmatia, ... | 55 | Croatian Locations |

Table 7: Example Retrieved Hash Clusters

