# OpenReview forum: "Semantic Hashing with Locality Sensitive Embeddings"
_ICLR.cc/2021/Conference — Reject_

### Official Review · AnonReviewer4 · 2020-10-15
**This paper proposes an angular-based metric which tries to pull similar objects together and dissimilar ones apart. Three different experiments, i.e., a toy example, an image retrieval task, and a co-occurrence modeling task, demonstrate the benefits of the learned embeddings.**

**Rating:** 4
**Confidence:** 5

**Review:**

+++Pros.
-----The idea learning embeddings via angular similarity is interesting and important for exact hashing retrievals; and the design of experiments on different domains to validate the proposed method is worth encouraging.

+++Cons.
-----This paper just devised the angular similarity, whose contribution is limited for ICLR.
-----There are some technical minors or typos, such as the binary cross entropy loss in Eq.~(5) (should be “+\beta(1-y_{ij})log(……)”).
-----There are several grammar minors, such as “…utilizes cosine similarity to as the crossing layer between the two halves…” in the beginning of Section 3, “We measure precision, recall and F1-score on with the data generating factions as the target.” in Section 4.1 above Table 2; and some confusing expressions, such as “in Table 4.1.” (there is no Table 4.1.).
-----Better replace beta with \beta, lambda with \lambda, and keep them identical in texts and figures. And better organize figures with subtitles.
-----At the end of each equation marked (1), …, (10), there should be appropriate punctuations, such as commas.
-----The authors concluded that “The learned representations show superior performance in the exact hashing retrieval setting”. But the results in Table 2 does not support it.
-----Besides, the authors mentioned that … a “word2hashes” model which is novel to the best of the authors’ knowledge …, and I do not agree with such statements.
-----Actually, I think Figures 4-8 and Table 5 should be regarded as the main texts; and if so, this paper exceeds the page limit.

+++Conclusions.
-----Based on the above pros and cons, I think this paper is interesting, but the contribution is limited; thus, I would make a REJECT recommendation.

+++Suggestions.
-----Need careful writing and presentation.

---

> ### Author Response · Authors · 2020-11-12
> **Rebuttal to R4: We believe "representations for exact collision only retrieval" problem hasn't been solved before.  Solutions here open doors to many novel applications, making for high impact potential**
>
> Thank you for your review and your time.
>
> We vehemently disagree that the contribution is incremental.  There are three aspects to the contributions of our work
> 1) Potential/Novelty
> To our knowledge the "exact hashing representations problem" has not been seriously studied before, nor promising results shown. This has many potential applications and thus we belive high impact.  These applications are not simply limited to replacing ANN datastructures with hashing in IR systems, although that is one such application.
>  1a) This methodology can be be used as a nonparametric clustering/factorization method which
> can help practitioners reason about large scale data -- discrete clusters/representations are generally more interpretable to humans.  The size of each cluster is determined by the data -- giving another advantage in applications for data understanding.
>  1b) Exact hashing representations have potential to be used as intermediate representations in larger models for solving ANN problems without the need of intermediate searches.  For example, a multi-task scenario where hashes are learned based on a similarity label in one task and other targets are learned conditioned on these hashes in the other tasks.  The summarization achieved by the hashes enables simpler layers downstream.
> 1c) Retrieval algorithms storing only hashes are considerably more memory efficient (1 bit vs 1 double is a factor of 64 savings of memory). Thus exact hashing representations can help in scaling applications for both large scale retrieval systems, or can allow retrieval systems to run more easily on lightweight devices.
> 2) Algorithmic Contributions
> While each of these is fairly simple, each contribution complements the others and the end result is high quality semantic hashes
> 2a) Angular Similarity - We believe section 3.3 demonstrates angular similarity is a natural choice for hashing applications and perhaps better than existing choices
> 2b) Concentration Param (K)- All methods perform better when introducing this parameter
> 2c) Hard Negative Samples - Negative mining is common practice, but we found it to be very important for the success of exact hashing methods.  We provide a simple strategy for negative mining which works well in both synthetic and OSCAR datasets.
> 2d) Multiple-Hash model - For data where items may not fall naturally into single categories, multiple hashes as in the OSCAR expt can be used for "factorization" type models
> 3) Problem Space Contributions
> 3a) Large number of groups - Prior work has focused small number of semantic groups for evaluation.  We needed to include the K param for baselines for them to perform well in this regime
> 3b) Noisy Label - Prior work has focused on data with supervised labels, used to create a block diagonal matrix as dataset.  It's not clear similarity learning is needed at all in clean-label case (why not use supervised learning, eg)
>
>
> We fixed many of the typos/etc. For the BCE typo, the sign of the first and second term of the BCE didn't match (they should both be negative as BCE is a Negative LL)
>
> Can you be more specific when you say word2hashes is not novel?  Do you mean exact hashing representations of words?  We are not aware of any work using exact hashing retrieval (ie **only exact collisions used in retrieval**) for words.  If you know of some relevant work, would you please share it?

---

> ### Author Response · Authors · 2020-11-17
> **Updates on theoretical justification**
>
> Please see sections 3.3 and A.2 for theoretical justifications for angular similarity.

---

### Official Review · AnonReviewer2 · 2020-10-28
**The authors propose an approach for learning embeddings such that approximate nearest neighbor search becomes an exact hashing problem.  The experiments show that their approach is promising.**

**Rating:** 6
**Confidence:** 4

**Review:**

########
Pros
########

- The paper is well written and easy to read

- The authors have explained the background, motivation and prior work quite comprehensively

- The idea of learning embeddings such that nearest neighbors have the same hash is interesting.
Because of this property, nearest neighbor search becomes a simple hash lookup.



########
Cons
########

- The authors claim, "We extend semantic hashing methods to problems with **substantial label noise**", but there doesn't appear to be any specific modeling to handle label noise in Section 3.

- In practice, for IR, can this method support a ranking of the items based on query similarity? For example, if for query q, documents A, B, C are relevant in that order (most relevant to least). It appears that this method would all assign the same hash to A, B, C and therefore, retrieve them, but not in any particular order?

- Experiments don't appear to be necessarily a fair comparison that can show the merits of this approach conclusively.


########################
Comments / queries
########################

Q1 Section 4.1,  Table 1 shows very low numbers for DHN. Is that a typo, given that DHN performs quite well in all the other experiments.

Q2. Section 4.1, Table 1: Can you also provide the numbers for LSE without false negatives for fair comparison with the baselines?

Q3. Section 4.2, Table 2: It appears that LSE is able to perform well and beats the baselines in many cases. Can you provide the statistical significance p values?

Q4. Section 4.3, Table 3: LSE has a substantially lesser number of clusters, making it difficult to isolate the merits of the method vs. outcome that is a result of just lesser clusters. E.g. Recall can be better just by returning a larger cluster, which is probably the case if the number of clusters are lesser. Can you provide a fairer comparison, for example, by lowering the number of clusters of the other methods or some other way?


########
Typos
########

Section 4.2: "100 images per category it test query set"

Section 4.2: "Results are shown in Table 4.1." --> Table 2?

---

> ### Author Response · Authors · 2020-11-12
> **Rebuttal for R2: Additional plot in Fig 1 to show fundamental difference of DHN/logistic based similarity, expanded grid search**
>
> Thank you for your review and your time.
>
> Models trained using Angular similarity have a stronger inductive bias (see Fig 1) and thus can more easily handle noisy data. Appropriate negative sampling is also important for handling label noise, see Appendix (Fig 4) which may give some intuition on why our method of negative sampling can help to distinguish strongly connected groups from weakly coupled groups.
>
> HR metric on the test set for the OSCAR experiment (Table 3) does do sorting by dense embedding.  In addition, Figure 3 in main text and Figures 8 and 9 in Appendix show that there is additional structure for the dense embeddings associated with a single hash.  The remaining metrics for the "exact hashing experiments" do not do any ranking with retrieved results.
>
> Could you please elaborate on what you mean that comparison isn't fair?
>
>
> Q1: Expt 2 (Images) is using Hamming Ranking evaluation, on clean label, small # of groups tasks.  This explains why DHN can perform well here.  Metrics in other experiments are not typos.  The new plot in Fig 1 shows that it saturates easily (meaning loss is minimized too easily) resulting in poor performance for exact hashing retrieval case
> Q2: This comparison can be see in Fig 2, 3rd panel -- we increased the size of the plots in the rebuttal revision
> Q3: For Expt 2 we have computed pvalues according to the popular Iman Daveport test (results where significant for all datasets)
> Q4: Our apologies, the number of clusters column had an error in the computation (we didn't filter non-singleton hashes for all methods.)  We've updated with the correct values (which are all similar except DHN.)  Please note we are also reporting Precision which penalizes models producing overly large clusters.

---

> ### Author Response · Authors · 2020-11-17
> **Study on Hard Negatives**
>
> Please see Section A.1 in appendix for a study on the impact of hard negatives.  We note that DHN model improves in the scenario when hard negatives are removed.

---

### Official Review · AnonReviewer5 · 2020-11-10
**Theory needs to be justified more and some questions on the experiments**

**Rating:** 4
**Confidence:** 4

**Review:**

The authors consider the problem of learning a hash function such that semantically similar elements have high collision probability.  They modify the approach Deep Hashing Networks (Zhu et al., 2016) with a new loss function. Rather than use a sigmoid based loss function, the authors argue that a loss function based on angular similarity and SimHash would be better. Specifically, they use the probability of SimHash collisions as a loss function. They then experimentally verify their method on synthetic data from a Stochastic Block Model distribution, image data (CIFAR-10 and ImageNet), and text data (OSCAR). They show improvements over related methods.

Overall, I found this paper to be incremental compared to previous work, such as (Zhu et al., 2016). The theoretical contributions are fairly weak. Why is the relation to SimHash useful? The authors need to better justify their choice of loss function. Additionally, how does the method compare to using a sigmoid loss but changing the temperature parameter so that the loss function doesn't saturate as quickly?

The authors do improve over related methods in the experiments. However it is not clear if this is due to the choice of loss function or the use of negative mining. The authors should also improve the clarity of how their metrics are defined. Is precision, recall simply based on the elements that collide together?

To summarize, I think this paper needs to better justify their use of loss function in theory and also perform ablation tests in the experiments before it can be accepted.

---

> ### Author Response · Authors · 2020-11-12
> **Rebuttal for R5 - Added  theoretical analysis of Angular similarity and it's strong connection to hashing**
>
> Thank you for your review and your time.
>
> We've added a section (3.3) demonstrating connections between various similarities and probabilistic statements about hashing collisions.  In addition, we show a connection between angular distance/SimHash and hamming distance.  While these results do not definitively show that angular similarity will necessarily outperform other methods, we believe they do show it is a natural choice when training models for hashing based representations, with better analytical properties than existing choices.
> We have also added an additional plot in Figure 1 showing that Angular distance does indeed induce a sharper topology to baseline distances.
> This figure also shows the logistic similarity.  As can be seen, the inverse temperature parameter for the logistic similarity would need to be near 10^-6 to achieve the same behavior as angular similarity near the saturation point, however as the range of cosine similarity is bounded between -1 and 1, the saturation point would not be achievable with such a temperature.  For an inner product based input, embeddings would need large norms (10^3 for each) in order to reach the saturation point -- which may be feasible but would require significant information to be contained in the norm of the embeddings.
>
> We vehemently disagree that the contribution is incremental.  There are three aspects to the contributions of our work
> 1) Potential/Novelty
> To our knowledge the "exact hashing representations problem" has not been seriously studied before, nor promising results shown. This has many potential applications and thus we belive high impact.  These applications are not simply limited to replacing ANN datastructures with hashing in IR systems, although that is one such application.
>  1a) This methodology can be be used as a nonparametric clustering/factorization method which
> can help practitioners reason about large scale data -- discrete clusters/representations are generally more interpretable to humans.  The size of each cluster is determined by the data -- giving another advantage in applications for data understanding.
>  1b) Exact hashing representations have potential to be used as intermediate representations in larger models for solving ANN problems without the need of intermediate searches.  For example, a multi-task scenario where hashes are learned based on a similarity label in one task and other targets are learned conditioned on these hashes in the other tasks.  The summarization achieved by the hashes enables simpler layers downstream.
> 1c) Retrieval algorithms storing only hashes are considerably more memory efficient (1 bit vs 1 double is a factor of 64 savings of memory). Thus exact hashing representations can help in scaling applications for both large scale retrieval systems, or can allow retrieval systems to run more easily on lightweight devices.
> 2) Algorithmic Contributions
> While each of these is fairly simple, each contribution complements the others and the end result is high quality semantic hashes
> 2a) Angular Similarity - We believe section 3.3 demonstrates angular similarity is a natural choice for hashing applications and perhaps better than existing choices
> 2b) Concentration Param (K) - All methods perform better when introducing this parameter
> 2c) Hard Negative Samples - Negative mining is common practice, but we found it to be very important for the success of exact hashing methods.  We provide a simple strategy for negative mining which works well in both synthetic and OSCAR datasets.
> 2d) Multiple-Hash model - For data where items may not fall naturally into single categories, multiple hashes as in the OSCAR expt can be used for "factorization" type models
> 3) Problem Space Contributions
> 3a) Large number of groups - Prior work has focused small number of semantic groups for evaluation.  We needed to include the K param for baselines for them to perform well in this regime
> 3b) Noisy Label - Prior work has focused on data with supervised labels, used to create a block diagonal matrix as dataset.  It's not clear similarity learning is needed at all in clean-label case (why not use supervised learning, eg)
>
> Precision and Recall are defined on the elements that collide together for the 1st and 3rd experiments (this is what we mean by exact hashing retrieval,) for the 2nd experiment we use Hamming ranking to give consistent metrics with regard to prior work.
>
> Some ablations may already be viewed in the tuning plots (specifically "no quantization loss" corresponds to \lambda=0, "no K param" corresponds to K=1) -- we included these so that such ablation questions and dependencies on hyperameters can be easily checked.
>
> We are running an additional tuning experiment where we modify the loss to only change the weight of hard negatives, rather than all negatives. This enables us to more clearly show the impact of hard negatives.  This is a rather expensive grid search; it will take several days.

---

> ### Author Response · Authors · 2020-11-17
> **Updates as requested**
>
> Please see Sections 3.3, A.2 for updates on the theoretical justifications of the model, and Section A.1 for the ablation study and impact of hard negatives.

---

### Author Response · Authors · 2020-11-12
**Paper Updates**

We have made an update to the paper with several revisions.  We have also included the "word2hashes" model hashes as a parquet file in the supplementary material.  Unfortunately the dense versions will not fit in the 100MB file limit

Major changes:
Added section 3.3 - theoretical analysis of Angular similarity as it relates to hashing
Additional plot in Fig 1 showing angular similarity induces the sharpest topology among alternatives and thus introduces the strongest preference for near-binary embeddings
Increased the range of the OSCAR hyperparameter search (baselines improve but do not surpass LSE model)

---

> ### Author Response · Authors · 2020-11-17
> **Update with Ablation Study**
>
> We have updated the paper to include a hyperparameter sweep which is modifying the weight of hard negatives, rather than all negatives.  We see that DHN performance improves considerably when hard negatives are removed.  We use this parameter sweep for an ablation study, showing the performance of each model under no hard negatives, K=1, and \lambda=0, respectively.  Please see Section A.1 in the appendix for details.

---

### Decision · Program_Chairs · 2021-01-07
**Final Decision**

**Decision:**

Reject

**Comment:**

Thanks for your submission to ICLR.

This paper presents an extension to Deep Hashing Networks that utilizes angular similarity, and show improved results using the proposed method.  The reviewers were somewhat mixed on this paper, with two of three reviewers on the negative side.  Some reviewers appreciated that the paper was easy to follow and well written, though one reviewer felt that the paper's writing and presentation could improve.  A big concern about the paper expressed by multiple reviewers was that the paper was incremental, in that the main architectural difference seemed to be a change in loss function over existing work.  Unfortunately, the reviewers were fairly unresponsive to attempts to get them to respond to the rebuttals offered by the authors.

Ultimately, I took a look at the paper and found it to be borderline.  I do think the contribution is a bit limited, particularly as it is in an area which has seen many papers over the years (and thus has a high bar for new work).  However, with some additional work this paper could definitely be acceptable.  I think it could use an additional round of editing and review, and I'd encourage the authors to submit this paper to another venue.